# ABKD: Graph Neural Network Compression with Attention-Based Knowledge Distillation

## Abstract

Graph Neural Networks (GNNs) have proven to be quite versatile for a variety of applications, including recommendation systems, fake news detection, drug discovery, and even computer vision. Due to the expanding size of graph-structured data, GNN models have also increased in complexity, leading to substantial latency issues. This is primarily attributed to the irregular structure of graph data and its access pattern into memory. The natural solution to reduce latency is to compress large GNNs into small GNNs. One way to do this is via knowledge distillation (KD). However, most KD approaches for GNNs only consider the outputs of the last layers and do not consider the outputs of the intermediate layers of the GNNs; these layers may contain important inductive biases indicated by the graph structure. To address this shortcoming, we propose a novel KD approach to GNN compression that we call Attention-Based Knowledge Distillation (ABKD). ABKD is a KD approach that uses attention to identify important intermediate teacher-student layer pairs and focuses on aligning their outputs. ABKD enables higher compression of GNNs with a smaller accuracy dropoff compared to existing KD approaches. On average, we achieve a $1.79\%$ increase in accuracy with a $32.3\times$ compression ratio on OGBN-Mag, a large graph dataset, compared to state-of-the-art approaches.

## 1 Introduction

Graph Neural Networks (GNNs) generalize Convolutional Neural Networks (CNNs) to non-Euclidean data. GNNs are widely used in a variety of fields, such as web-search recommendation systems (1), fake news detection for social networks (2), modeling proteins for drug discovery (3), and computer vision tasks (4). Due to the expanding size of social networks and other graph-structured data, graph datasets have been steadily increasing in size (5). As datasets have expanded in size, GNN models have also increased in complexity, leading to substantial latency issues (6; 7; 8), as shown in figure 1. This is primarily attributed to the irregular structure of graph data and their access pattern in memory (9).

Due to this limitation, large GNNs need to be compressed into smaller GNNs for latency-sensitive applications such as real-time recommendation (11), visual question answering (12), image search (13), and real-time spam detection (14).

### 1.1 Knowledge Distillation

Knowledge Distillation (KD) is a common compression technique that uses a teacher model to supervise the training of a smaller student model (15). While the original KD method can be applied to GNNs, it does not take into account any information about node connectivity. GraphAKD (16), LSP (17), and G-CRD (18) are GNN-specific knowledge distillation methods that focus on aligning final layer node embeddings by considering node connectivity. However, these methods all consider only the node embeddings

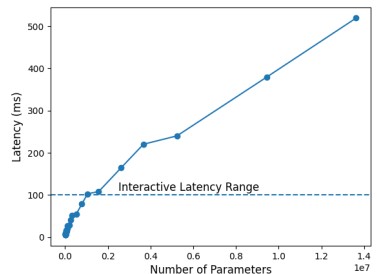

Figure 1: Inference latency of GNNs with varying model sizes on the Flickr (10) dataset on a standard GCN model architecture with increasing embedding dimension. All tests were run on a Tesla V100 GPU.

at the final layer and do not consider intermediate representations. By just aligning the final node embeddings, the model cannot learn the logic behind leveraging the connectivity of the graph or the inductive biases (19) contained in the adjacency matrix. Therefore, the student model is effectively just learning a mapping from node attributes to more refined node embeddings. This can lead to suboptimal test generalization when the model encounters previously unseen data.

**Objective**: Our goal is to achieve better generalization on out-of-distribution data, measured by accuracy @ various compression rates, by considering intermediate node embeddings and taking into account more of the inductive biases that GNNs contain.

## 1.2 ATTENTION-BASED KNOWLEDGE DISTILLATION

To improve accuracy over SOTA, we must consider some of the inductive biases that other methods overlook - the connectivity of the input graph. We observe that for the vast majority of GNN architectures, the $k^{th}$ layer of a GNN computes node embeddings by aggregating information from each node's $k$-hop neighborhood. Therefore, each layer contains its own inductive bias. As other KD methods consider only the node embeddings at the last layer of the teacher and student networks, they do not leverage all the inductive biases present in the network. However, one obvious challenge that is present in aligning intermediate node embeddings is that teacher and student networks will likely have a different number of hidden layers. As a result, there is no 1-1 correspondence between teacher and student layers and no way to easily figure out which teacher node embeddings should be aligned with which student node embeddings.

To tackle these challenges, we propose Attention-Based Knowledge Distillation (`ABKD`). We use a trainable attention mechanism to learn which teacher-student pairs are the most important to align. We also utilize trainable projections into a common `ABKD` embedding space for teacher and student hidden layers. By aligning across intermediate layers, the student learns how to use the adjacency matrix to construct node embeddings instead of just learning a mapping from the node attributes to the final layer node embeddings (Figure 2).

## 1.3 OUR CONTRIBUTIONS

Our contributions are summarized as the following:

1. We design Attention-Based Knowledge Distillation (`ABKD`), a novel knowledge distillation approach for GNNs that incorporates the intermediate feature maps from every layer in both the teacher and student networks. This approach can be utilized to train teacher and student networks of any architectural configuration.

2. We create an automatic feature linking mechanism using attention to identify which teacher-student layer pairs are the most important, which we then use to closely align their feature maps.

3. Our approach broadly improves the test accuracy of student networks over a large range of compression ratios.

4. We comprehensively test our approach on several datasets using different model architectures including GCNs (20), RGCNs (21) and GraphSAGE (22). We also test on several large datasets that are carefully curated to evaluate out-of-distribution generalization such as OGBN-Mag (5) and OGBN-Arxiv (5).

## 2 RELATED WORK

**Knowledge Distillation** KD for GNNs is a relatively niche field that has expanded over the last three years with the work of LSP (17). In this work, the authors attempt to align node embeddings between the student and teacher networks by maximizing the similarity between embeddings that share edges. As only node embeddings between edges are aligned, this KD method only preserves local topology. Joshi et. al (18) extend LSP and propose two different KD algorithms: Global Structure Preserving Distillation (GSP) and Global Contrastive Representation Distillation (G-CRD). GSP extends LSP by considering all pairwise similarities among node features, not just pairwise similarities between nodes connected by edges. The authors also propose G-CRD which aims to implicitly preserve

global topology by aligning the student and teacher node feature vectors via contrastive learning (23). Another work introduces graph adversarial knowledge distillation (GraphAKD), which trains the student model as a generator network and a discriminator network to distinguish between the predictions of the student and teacher models (16). Another work, GeometricKD forces the number of teacher and student hidden layers to be the same to study the impact of a student network operating on a smaller graph than the teacher (24).

With the exception of GeometricKD, these works all consider the node embeddings at the final layer of the teacher and student network and aim to align those embeddings with one another in various ways. GeometricKD constrains the student and teacher networks to have the same number of layers, and it aligns the node embedding of teacher layer $i$ with student layer $i$, thus forming a 1-1 correspondence between the layers, which makes it inflexible to all student and teacher configurations. There have been several KD approaches that have been applied to CNNs that the GNN community has tried to adapt to GNNs with poor results, namely Fitnets (25) and attention transfer (26). These methods both compute a distance metric such as mean-squared error between the last layer node embeddings of the student and teacher network and do not take into account the adjacency matrix.

Using attention to find similarities across student and teacher layers is a concept explored in CNNs (27). However, this work's ideas cannot be applied to GNNs because the operations it uses to compare student and teacher features do not apply to GNNs. GNNs need special consideration in this regard over CNNs due to the non-spatial and unstructured form of graph data.

**Compression:** Other common techniques for GNN compression include quantization and pruning. Quantization techniques differ for GNNs when compared to other deep neural networks because of unique sources of error, such as inaccurate gradient estimates, that arise due to the unstructured nature of GNNs (28). To account for inaccurate gradient estimates, DegreeQuant, protects some number of nodes in every training iteration from lower precision training and utilizes full precision training for those nodes (28). Other approaches consider the structure of the graph in their calculations to implicitly avoid GNN-specific sources of error (29; 30).

Pruning techniques, which involve compressing networks by selectively deleting learned parameters, have also been applied to GNNs. These techniques involve applying strategies that have worked for other deep networks and using them to identify the most important channels for output activation (31) (32). One approach also works to dynamically prune during execution (33).

| Method | Number of Layers Considered |
|---|---|
| GraphAKD | 1 |
| G-CRD | 1 |
| Fitnets | 1 |
| LSP | 1 |
| ABKD | **All** |

Table 1: Comparison of Attention-Based Knowledge Distillation with other Knowledge Distillation approaches.

## 3    PROPOSED APPROACH

### 3.1    INTUITION AND MATHEMATICAL FOUNDATIONS

In this section, we first discuss the intuition behind ABKD and introduce some of the mathematical definitions needed to explain it thoroughly.

#### 3.1.1    SOFTKD INTUITION

In SoftKD (15), we compute two different losses. The first, $H(s_p, y)$, is a cross-entropy loss between the output student probability distribution and the ground truth labels. The other, $H(s_p, t_p)$, is a cross-entropy loss between the output student probability distribution and the output teacher probability distribution. The total loss is defined as:

$$L_{KD} = H(s_p, y) + \alpha H(s_p, t_p) \tag{1}$$

Here, $\alpha$ is a hyper-parameter controlling how much the KD loss affects the total loss. The goal is to align the output student probability distribution with the output teacher probability distribution. The higher $H(s_p, t_p)$ is, the less aligned the student and teacher output probability distributions are.

### 3.1.2  ABKD INTUITION

Similarly, with ABKD, we want to incorporate this intuition of alignment. However, we want to go further than just aligning the final output - we want to align the outputs at the intermediate layers, as the intermediate layers contain inductive biases that are not present in just the final layers. As one of our goals with ABKD is to work with any combination of teacher and student architectural configurations, this presents one significant challenge: Teacher and student networks will likely have a different number of hidden layers, which means there is no 1-1 correspondence between teacher and student layers.

ABKD solves this problem by identifying which teacher-student layer pairs are the most important to align via an attention mechanism. This mechanism works with any arbitrary number of teacher layers and student layers, which makes this approach amenable to any arbitrary teacher-student configuration. ABKD also uses a reprojection technique to account for the student and teacher networks having different hidden dimensions. The output of each hidden layer for both the teacher and student networks is projected into a standardized embedding dimension, which ensures that we can work with student and teacher networks of any embedding dimension. As each layer represents its own semantic information, an important challenge that we faced was to ensure that each layer's feature map was not smoothed out by a single projection matrix. To this end, we use separate trainable linear layers for each hidden layer in both the teacher and student networks, in order to ensure that we don't lose out on any valuable semantic information in the hidden layers.

These trainable linear layers help us construct the two key components of ABKD, which are the attention map and the dissimilarity map. At a high level, the attention map tells us how important each teacher-student layer pair is, while the dissimilarity map tells us how distant the feature maps of each teacher-student layer pair are. The teacher-student layer pairs with higher attention scores are deemed as more important, and ABKD focuses on reducing their dissimilarity scores during training.

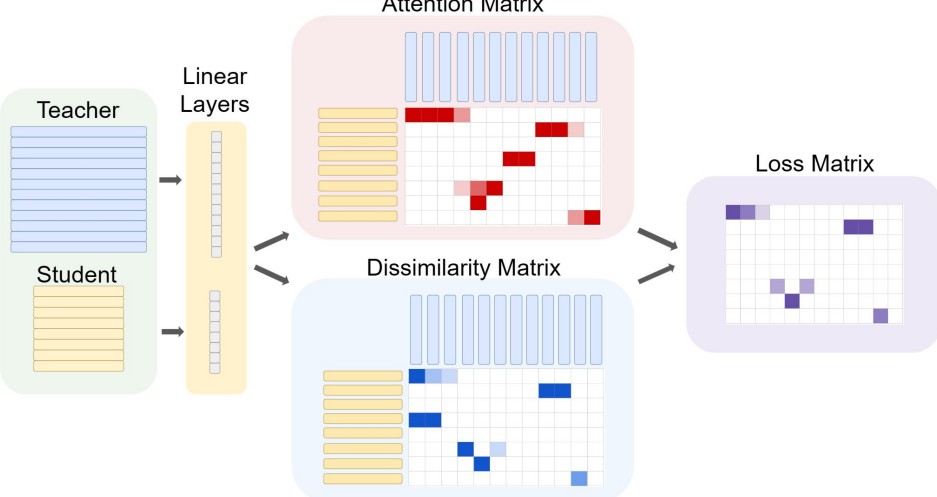

Figure 2: ABKD generates an attention map using a trainable attention mechanism and a dissimilarity map using a trainable subspace projection. The loss matrix is an element-wise multiplication of the attention matrix and the dissimilarity matrix.

### 3.1.3  MATHEMATICAL FOUNDATION

Without loss of generality, we will consider distilling a general Graph Convolution Network (GCN) (20), in which the output of the $l^{th}$ layer is

$$H^l = \sigma(\hat{A}H_{l-1}W_l) \tag{2}$$

Here, $\sigma$ is an activation function and $\hat{A}$ is the normalized adjacency matrix, $\hat{A} = D^{-\frac{1}{2}} A D^{-\frac{1}{2}}$, where $D$ is the diagonal degree matrix. $H^{l-1} \in R^{n \times d}$ represents the output of the last hidden layer and $W^l \in R^{d \times d}$ represents the trainable weights of the current layer.

Consider two different tensors $T \in R^{T_l \times n \times d_t}$ and $S \in R^{S_l \times n \times d_s}$. $T_l$ and $S_l$ represent the number of layers in the teacher and student networks respectively. $n$ represents the number of nodes in the graph that is being trained on and $d_t$ and $d_s$ represents the dimensionality of the teacher and student networks, respectively. $T$ and $S$ represent the calculated layer maps before activation is applied at every layer for the teacher and student networks, respectively.

## 3.2 ABKD

### 3.2.1 ATTENTION SCORES

The first step of ABKD is to generate $A \in R^{T_l \times S_l}$. $A_{ij}$ will represent an "importance" score for the layer pair consisting of teacher layer $i$ and student layer $j$. We take the average of the feature maps along the node dimension to compute a mean node feature for every layer in both the teacher and student networks. Call these tensors $T_a \in R^{T_l \times d_t}$ and $S_a \in R^{S_l \times d_s}$. Then, we pass each layer in $T_a$ through its own linear layer to create $T_p \in R^{T_l \times d_a}$, where $d_a$ is the embedding dimension of ABKD. Similarly, we create $S_p \in R^{S_l \times d_a}$. We can finally generate $A$ in the following manner:

$$A = softmax(\frac{T_p S_p^T}{\sqrt{d_a}}) \tag{3}$$

### 3.2.2 DISSIMILARITY SCORES

The next step is to compute a pairwise dissimilarity score for each teacher-student layer pair. Again, we project the features into $d_a$. For calculating the attention scores, we averaged over the node dimension before projecting, as our goal was to identify important layers. When calculating the pairwise dissimilarity, we want to incorporate the per-node embeddings. So, we use a separate set of projection matrices. We use $P_t \in R^{d_t \times d_a}$ and $P_s \in R^{d_s \times d_a}$ to represent the projections.

However, distance metrics are less semantically valuable if $d_a$ is high. To alleviate this problem, we define a trainable matrix $P \in R^{d_a \times d_a}$ to project all vectors into the subspace defined by the column space of $P$. Since the cardinality of the subspace defined by the column space of $P$ will be smaller than or equal to the cardinality of $R^{d_a}$, distance metrics within the subspace will be more valuable on average compared to distance metrics in $R^{d_a}$.

The final step is to average over the embedding dimension and then produce $D \in R^{T_l \times S_l}$, which gives dissimilarity scores for each teacher-student layer pair. For calculating the dissimilarity, we experiment with Euclidean and cosine distance, but Euclidean distance generally tends to perform better. The dissimilarity score for a layer pair $(i, j)$ can be represented as:

$$D_{ij} = ||(T_i P_t - S_j P_s) P \frac{\mathbb{1}_{d_a}}{d_a}||_2^2 \tag{4}$$

where $\mathbb{1}_{d_a}$ is a vector of 1s in $R^{d_a}$.

### 3.2.3 FINAL LOSS CALCULATION

To produce the final loss matrix, we element-wise multiply $A$ and $D$ and then take the row-wise mean to produce a single number that represents the ABKD loss.

$$L_{abkd} = (\mathbb{1}_{T_l})^T (A \odot D)(\frac{\mathbb{1}_{S_l}}{S_l}) \tag{5}$$

The final loss is calculated as

$$L = H(s_p, y) + \beta L_{abkd} \tag{6}$$

There is one important theorem to consider that proves $L_{abkd}$ distills valuable knowledge from the teacher network to the student network.

**Theorem 1:** Consider a teacher layer $i$ and a student layer $j$ such that $j < i$. Consider the weight $W_j^s$ which is the associated weight for the $j^{th}$ layer of the student network. The weight update for $W_j^s$ will be affected by $W_i^t$, which are the weights for the $i^{th}$ layer of the teacher network.

**Proof:** To calculate the weight update, we have to first formulate a loss for the layer pair $(i, j)$. By adapting equation 5 to a specific element in the loss matrix, we get:

$$L_{ij} \sim (\frac{\mathbb{1}_n}{n})^T (T_i^p (W_j^{ps})^T)(W_j^s)^T ((H_{j-1}^s)^T \hat{A}^T)) \frac{\mathbb{1}_n}{n} * D_{ij}$$
$$T_i^p = \hat{A} H_{i-1}^t W_i^t W_i^{pt}$$

where $W_i^{pt}$ and $W_j^{ps}$ represent the projection matrices used to calculate the attention score. Assuming that our weight updates are performed by gradient descent, to calculate the weight update for $W_j^s$ due to $L_{ij}$, we have to first calculate $\frac{\partial L_{ij}}{\partial W_j^s} \in R^{d_s \times d_s}$:

$$\frac{\partial L_{ij}}{\partial W_j^s} = \frac{\partial A_{ij}}{W_j^s} D_{ij} + \frac{\partial D_{ij}}{W_j^s} A_{ij} \tag{7}$$

We focus on the first term and obtain the result:

$$\frac{\partial A_{ij}}{\partial W_j^s} \sim ((H_{j-1}^s)^T \hat{A}^T (\frac{\mathbb{1}_n}{n}))((\frac{\mathbb{1}_n}{n})^T T_i^p (W_j^{ps})^T) \tag{8}$$

It is apparent that as $\frac{\partial A_{ij}}{\partial W_j^s}$ contains the term $T_i^p = \hat{A} H_{i-1}^t W_i^t W_i^{pt}$, the weight update will reflect terms from the $i^{th}$ teacher layer. This theorem goes to show that even though student layer $j$ is responsible for aggregating information for every node from its $j$-hop neighborhood, its weights collect knowledge from teacher layer $i$ - knowledge that would be unavailable to student layer $j$ using any of the other GNN KD approaches.

An important follow-up observation is that the loss for a teacher-student layer pair will be higher if the pair is deemed as more important and if their projected feature maps have a high dissimilarity score ($D_{ij} > 0.5$), which indicates that they aren't closely aligned.

To verify this observation, consider an arbitrary teacher layer $i$ and an arbitrary student layer $j$. Consider $A_{ij}$. If the pair $(i, j)$ is deemed important, $A_{ij}$ will be high. Now consider $D_{ij}$. If the pair $(i, j)$ is not closely aligned, then $D_{ij}$ will be high. As $L_{ij} = A_{ij} * D_{ij}$, if both $A_{ij}$ and $D_{ij}$ are high, then $L_{ij}$ will also be high, hence proving that the loss is high for important but misaligned layer pairs.

## 4 EXPERIMENTS

### 4.1 EXPERIMENTAL SETUP

For our main experiments, we test ABKD on two difficult datasets OGBN-Mag and OGBN-Arxiv (5). These datasets employ temporal splitting to create validation and test sets that can assess a model's ability to generalize on out-of-distribution data (34). For OGBN-Mag, we run experiments using RGCN (21) as the teacher and student models, and for OGBN-Arxiv, we run experiments using GAT (35) as the teacher model and GraphSAGE (22) as the student model. This allows us to evaluate the effectiveness of ABKD for different GNN architectures. It also allows us to assess if ABKD can distill information between different types of GNN architectures. In our experiments, we keep the teacher model architecture and weights fixed and only modify the size of the student network. Each distillation method starts from the same set of weights and trains for the same number of epochs across 5 runs. For our baselines, we consider LSP (17), GSP (18), G-CRD (18), Fitnet (25), and Attention Transfer (26). We ran all experiments on a Tesla V100 GPU.

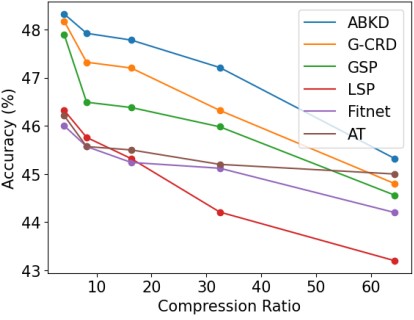

Figure 3: Accuracy vs. Compression Ratio: A comparison of KD methods applied to student models of different sizes trained on the OGBN-Mag dataset. The teacher model was the same model as the one described in table 3. The student model was a two-layer RGCN and we varied the embedding dimension from 16 - 512 to induce this Pareto frontier. The y-axis represents the test accuracy of the final trained models.

| Dataset | OGBN-Mag | OGBN-Arxiv |
|---|---|---|
| Teacher | RGCN (3L-512H-5.5M) | GAT (3L-750H-1.4M) |
| Student | RGCN (2L-32H-170K) | GS (2L-256H-87K) |
| Teacher | 49.80 | 74.20 |
| Student | $44.23 \pm 0.47$ | $70.87 \pm 0.58$ |
| Fitnet | $44.87 \pm 0.84$ | $71.32 \pm 0.32$ |
| AT | $43.87 \pm 0.67$ | $71.04 \pm 0.48$ |
| LSP | $45.21 \pm 0.54$ | $71.47 \pm 0.45$ |
| GSP | $44.97 \pm 0.58$ | $71.97 \pm 0.64$ |
| G-CRD | $45.42 \pm 0.43$ | $71.87 \pm 0.56$ |
| ABKD | $\mathbf{47.21 \pm 0.32}$ | $\mathbf{73.25 \pm 0.56}$ |
| Ratio | $32.3\times$ | $16.1\times$ |

Table 2: Average accuracies over five trials for the OGBN-Mag and OGBN-Arxiv Datasets. The teacher network was kept constant while the different distillation methods were applied to student networks that were initialized from the same set of weights. GS means GraphSAGE.

## 4.2 EXPERIMENTAL RESULTS

**Out-Of-Distribution Evaluation** As evidenced by our results in Table 2, we are able to outperform SOTA in several different compression settings on OGBN-Mag and OGBN-Arxiv. We show improvements of over $1.79\%$ and $1.38\%$ with compression ratios of $32.3\times$ and $16.1\times$ on OGBN-Mag and OGBN-Arxiv respectively. As these datasets are intended to evaluate out-of-distribution generalization, our results empirically prove that ABKD can train student models that generalize better than other KD approaches.

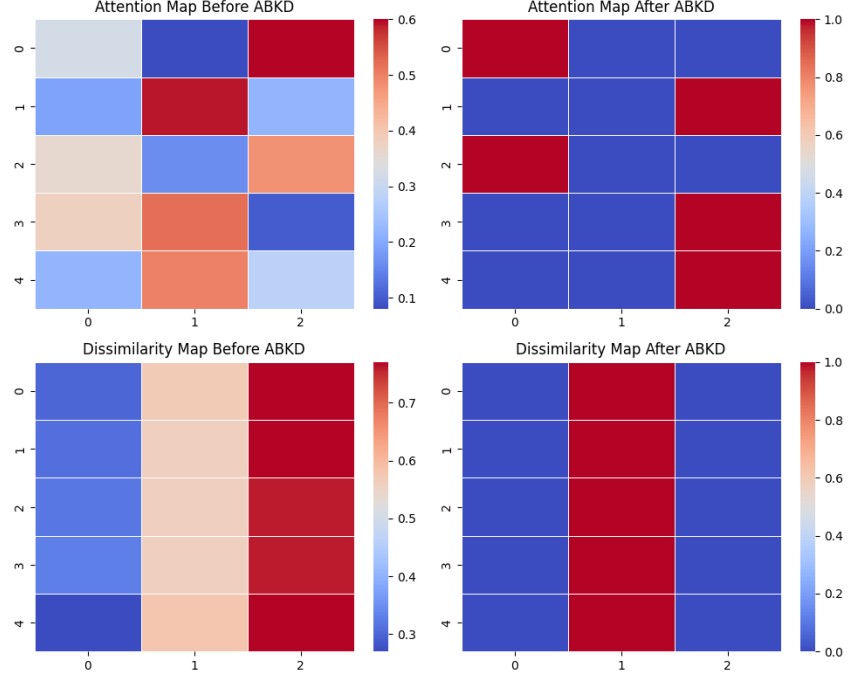

Figure 4: Attention and Dissimilarity maps before and after training with ABKD. Cooler colors refer to lower scores and warmer colors correspond to higher scores.

**ABKD aligns intermediate embeddings**: To empirically prove that ABKD actually aligns intermediate embeddings based on the attention matrix, we visualize the before and after training attention and dissimilarity maps in figure 4. We train on OGBN-Mag and use a deeper teacher network of 5 layers and a hidden dimension of 512. The student network has 3 layers and a hidden dimension of 32. Our results show that dissimilarity scores are low where the attention scores are high and vice versa. This is in line with the intuition that we constructed in 3.2.3.

| Dataset | Cora | Citeseer | Pubmed | NELL |
|---|---|---|---|---|
| **Teacher** | GCNII (64L-64H) | GCNII (64L-64H) | GCNII (64L-64H) | GCNII (64L-64H) |
| **Student** | GCNII (4L-4H) | GCNII (4L-4H) | GCNII (4L-4H) | GCNII (4L-4H) |
| Teacher | 88.40 | 77.33 | 89.78 | 95.55 |
| Student | $74.22 \pm 1.32$ | $68.14 \pm 1.45$ | $88.66 \pm 0.87$ | $85.20 \pm 0.75$ |
| LSP | $75.18 \pm 1.45$ | $70.43 \pm 1.32$ | $89.01 \pm 1.73$ | $85.20 \pm 1.32$ |
| GSP | $78.32 \pm 1.21$ | $69.43 \pm 0.87$ | $89.13 \pm 1.05$ | $86.13 \pm 1.47$ |
| G-CRD | $83.35 \pm 1.32$ | $71.42 \pm 1.21$ | $89.73 \pm 1.05$ | $88.61 \pm 1.01$ |
| ABKD | $\mathbf{84.27 \pm 1.48}$ | $\mathbf{71.96 \pm 0.87}$ | $\mathbf{89.87 \pm 0.45}$ | $\mathbf{91.83 \pm 0.74}$ |
| # Parameters | 5835 | 14910 | 2083 | 22686 |
| Ratio | $60.7\times$ | $33.5\times$ | $141.3\times$ | $27.7\times$ |

Table 3: Average accuracies over several trials for a variety of datasets, using GCNII. The results are based on the average of five trials, with each distillation method applied to the same set of student weights.

**Deep GNNs**: We also test ABKD on deep GNN architectures. While most GNN architectures are shallow due to the problem of over-smoothing, there are some approaches that alleviate this issue, allowing for deep GNN architectures. One of these approaches is GCNII (36). We test on Cora (37), Citeseer (38), Pubmed (39) and NELL (40). Table 3 shows that ABKD is able to distill these deep GCNIIs into shallower GCNIIs with higher accuracy compared to other distillation methods in high compression settings. While currently, most GNN use cases require shallow GNNs, this could be useful for future applications that require deeper GNNs.

**Improved Weight Initialization for Highly Compressed Networks**: We find that for smaller datasets, information from the teacher network is mainly distilled into one layer of the student network, as shown in Figure 5. Our hypothesis for this occurrence is that smaller datasets are not very complex and one layer is sufficient for learning most of the patterns. Through experimentation, we prove that when we initialize a one-layer network from the weights of this particular layer, we get improved accuracy compared to random initialization, as shown in Table 4.

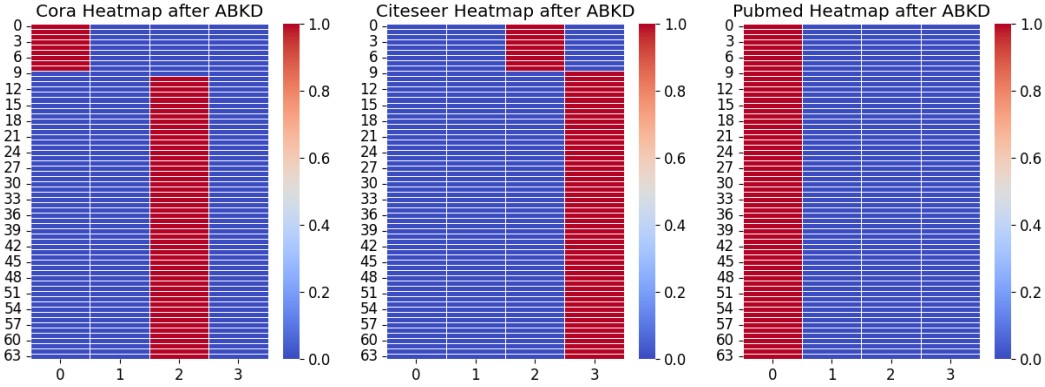

Figure 5: Attention maps for Cora, Citeseer, and Pubmed. Each color in the heatmap represents the importance score associated with that teacher-student layer pair. Warmer colors mean higher important scores. It seems apparent that most of the knowledge from the teacher layers is distilled into one student layer.

| Dataset | Initialized | Random Init | No Training |
|---|---|---|---|
| Cora | 80.29 | 73.58 | 65.36 |
| Citeseer | 70.12 | 68.12 | 54.20 |
| Pubmed | 88.76 | 86.03 | 72.90 |

Table 4: Results for weight initialization experiment. Experimental details can be found in the appendix.

| Dataset | OGBN-Mag | OGBN-Arxiv |
|---|---|---|
| Student | $44.46 \pm 0.54$ | $71.27 \pm 0.48$ |
| ABKD | $\mathbf{47.46 \pm 0.45}$ | $\mathbf{73.18 \pm 0.56}$ |
| Modified ABKD | $46.56 \pm 0.60$ | $71.89 \pm 0.87$ |

Table 5: Ablation study comparing modified ABKD that only considers aligning the last layer node embeddings with ABKD that considers intermediate layer node embeddings. The teacher and student models are the same as the ones in Table 2.

## 4.3 ABLATION STUDIES

**Aligning Intermediate Layers is Important**: To prove that the aligning of intermediate layers is necessary for superior performance, we experiment with a variant of ABKD, which we call modified ABKD, where we set $A \in R^{T_l \times S_l}$ to all zeros, but we set the bottom right value to 1. This indicates that we are only interested in the dissimilarity between the last layer node embeddings of the teacher and student models. The results in table 5, prove that we gain accuracy by considering the outputs of intermediate layers for both teacher and student models. In this experiment, we start from the same set of initialized weights for both the ABKD and modified ABKD approaches.

**Improvements due to Subspace Projection**: In our approach, we introduced the concept of subspace projection as an alleviation to high dimensional embedding spaces. While it is not needed for ABKD to work, it does improve our results as the learned subspace projection matrix tends to be of lower rank than the embedding dimension. This indicates that we can project our feature maps into subspaces smaller than $R^{d_a}$, which increases the semantic value of the dissimilarity scores.

**One Linear Layer Per Hidden Layer Is Necessary**: In our approach, we mention that each hidden teacher and student layer is assigned its own linear layer for projection into $d_a$. This is because each layer represents its own $k$-hop neighborhood, and using just one linear layer would prove inadequate in capturing the full spectrum of essential semantic information contained within each layer. We run an experiment in which we use only one linear layer for the teacher and student projections and demonstrate that there is a significant accuracy dropoff compared to using individual linear layers for the projection.

| Dataset | Subspace Projection | No Projection |
|---|---|---|
| Cora | $84.27 \pm 1.32$ | $83.78 \pm 1.25$ |
| Citeseer | $71.96 \pm 0.83$ | $71.46 \pm 1.21$ |
| Pubmed | $89.87 \pm 0.87$ | $89.03 \pm 1.04$ |
| NELL | $91.83 \pm 0.85$ | $90.98 \pm 0.75$ |
| OGBN-Mag | $47.21 \pm 0.32$ | $46.83 \pm 0.45$ |
| OGBN-Arxiv | $73.25 \pm 0.56$ | $72.87 \pm 0.34$ |

Table 6: Subspace Projection Ablation Results. The teacher and student networks are the same as the ones described in tables 2 and 3.

| Dataset | Multiple Linear Layers | One Linear Layer |
|---|---|---|
| Cora | $84.27 \pm 1.48$ | $75.52 \pm 1.32$ |
| Citeseer | $71.96 \pm 0.87$ | $68.86 \pm 1.27$ |
| Pubmed | $89.87 \pm 0.45$ | $88.75 \pm 0.64$ |
| NELL | $91.83 \pm 0.74$ | $85.38 \pm 0.68$ |
| OGBN-Mag | $47.21 \pm 0.32$ | $44.67 \pm 0.54$ |
| OGBN-Arxiv | $73.25 \pm 0.56$ | $71.23 \pm 0.47$ |

Table 7: Linear Layer Ablation Results. The teacher and student networks are the same as the ones described in tables 2 and 3.

## 5 CONCLUSION

Graph Neural Networks (GNNs) have increased in complexity in recent years, owing to the rapidly increasing size of graph datasets. This increase in complexity poses a problem for latency-sensitive applications such as real-time recommendation and real-time spam detection, as GNN model sizes do not scale well with latency. To address this predicament, we propose an innovative solution known as Attention-Based Knowledge Distillation (ABKD). ABKD employs an attention-based feature linking mechanism to identify important intermediate teacher-student layer pairs and focuses on aligning the node embeddings of those pairs. This knowledge distillation approach broadly outperforms SOTA GNN-specific KD approaches over a wide variety of compression settings. It also works with both deep and shallow networks as proven by our experiments and performs well with several different types of GNN architectures. On average, we achieve a $1.79\%$ increase in accuracy with a $32.3\times$ compression ratio on OGBN-Mag, a large graph dataset, compared to state-of-the-art approaches.

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
