## SUPPLEMENTARY MATERIALS

## A CODE RELEASE

Our codebase can be found at `https://anonymous.4open.science/r/abkd_gnn-F658/`.

## B SUMMARY OF NOTATIONS

In Section 3, we mathematically describe how ABKD generates the attention matrix, $A$, and the dissimilarity matrix, $D$, which is then used to calculate $L_{\text{ABKD}}$. In Table 8, we provide a summary of all the mathematical notation used to describe ABKD.

| Term | Definition |
|---|---|
| $A_{ij}$ | Attention score for the teacher-student layer pair $(i, j)$ |
| $D_{ij}$ | Dissimilarity score for the teacher-student layer pair $(i, j)$ |
| $\mathbb{1}_n \in R^n$ | Vector of ones in $R^n$ |
| $T_i \in R^{n \times d_t}$ | The output of the $i^{th}$ layer in the teacher network. |
| $S_j \in R^{n \times d_s}$ | The output of the $j^{th}$ layer in the student network |
| $T_i^p \in R^{d_a}$ | The projected output of the $i^{th}$ student layer after taking the node-wise mean |
| $W_i^{pt} \in R^{d_t \times d_a}$ | The learnable $i^{th}$ projection matrix for the teacher network |
| $W_j^{ps} \in R^{d_s \times d_a}$ | The learnable $j^{th}$ projection matrix for the student network |
| $P_t \in R^{d_t \times d_a}$ | The learnable projection matrix for the teacher network; used to calculate the dissimilarity score |
| $P_s \in R^{d_s \times d_a}$ | The learnable projection matrix for the student network; used to calculate the dissimilarity score |
| $P \in R^{d_a \times d_a}$ | The learnable subspace projection matrix. Shared between the student and teacher networks |

Table 8: Summary of mathematical notation used in Section 3.

## C FURTHER EXPERIMENTAL DISCUSSION AND ABLATIONS

### C.1 DATASET AND TEACHER NETWORK INFORMATION

Tables 6 and 7 provide information on the datasets used in Section 4 and the corresponding teacher networks employed to supervise the training of various student networks during our experiments.

| | # of Nodes | # of Edges | # of Features | # of Classes |
|---|---|---|---|---|
| Cora | 2,708 | 10,556 | 1,433 | 7 |
| Citeseer | 3,327 | 9,104 | 3,703 | 6 |
| Pubmed | 19,717 | 88,648 | 500 | 3 |
| OGBN-Mag | 1,939,743 | 21,111,007 | 128 | 349 |
| OGBN-Arxiv | 169,343 | 1,166,243 | 128 | 40 |

Table 9: Specification of evaluated datasets.

### C.2 WEIGHT INITIALIZATION EXPERIMENT DETAILS

In Section 4.2, we present the results of an experiment in which we instantiate a one-layer network from the attention maps generated by ABKD. The first step of this experiment is to run ABKD on a student network of any arbitrary size and then generate the attention map, $A \in R^{T_l \times S_l}$. The next step is to use a row-wise argmax and find the student layer that has the most information distilled down to it. For example, in Figure 5, the selected layer for Cora would be the third student layer (index 2 in the figure). We then proceed to instantiate a new one-layer network and copy over the weights from the identified layer from $A$. We then evaluate this new network on the test set and report the results in column 3 of Table 4. The first column of Table 4 represents the accuracies that we obtain after we train the new one-layer network for 1200 epochs; we compare this result to the accuracy obtained from training a one-layer network from random initialization, which we report in the second column of Table 4.

## C.3 HYPERPARAMETER ABLATION STUDIES

There are two main hyperparameters that need to be tuned when training `ABKD`: the loss coefficient, $\beta$, and the `ABKD` embedding dimension, $d_a$. We present the test accuracies across various values for $\beta$ and $d_a$ in Tables 10 and 11. These results show that a lower $\beta$ and a higher $d_a$ tend to produce slightly better results. For all of our experiments, we used a $\beta$ of 10 and a $d_a$ of 256 for this reason.

| Dataset | $\beta = 1$ | $\beta = 10$ | $\beta = 20$ | $\beta = 50$ |
|---|---|---|---|---|
| Cora | 86.92 | 84.71 | 86.19 | 85.64 |
| Citeseer | 73.20 | 74.33 | 71.82 | 69.82 |
| Pubmed | 89.03 | 89.97 | 88.32 | 88.56 |
| NELL | 90.86 | 88.73 | 90.14 | 91.32 |

Table 10: Ablation results for $\beta$.

| Dataset | $d_a = 64$ | $d_a = 128$ | $d_a = 256$ | $d_a = 512$ |
|---|---|---|---|---|
| Cora | 87.45 | 87.11 | 86.92 | 86.37 |
| Citeseer | 72.07 | 72.52 | 73.12 | 74.62 |
| Pubmed | 89.58 | 89.58 | 89.33 | 89.12 |
| NELL | 89.73 | 90.12 | 90.73 | 91.14 |

Table 11: Ablation results for $d_a$.

| Dataset | Euclidean Distance | Cosine Distance |
|---|---|---|
| Cora | 87.33 | 82.81 |
| Citeseer | 73.43 | 70.98 |
| Pubmed | 89.58 | 87.32 |
| NELL | 91.24 | 85.66 |

Table 12: Test accuracies when using Euclidean distance vs. cosine distance for computing $D$.

In Section 3.2.2, we mention that we use a Euclidean distance metric instead of a cosine distance metric to generate the dissimilarity matrix, $D$. We present the results of this ablation in Table 12. As cosine distance disregards the magnitude of the vectors, we hypothesize that the magnitude of the hidden layer outputs is important. Furthermore, as Euclidean distance takes into account the magnitude of the hidden layer outputs, this is likely why it performs significantly better than cosine distance.