# OpenReview forum: "ABKD: Graph Neural Network Compression with Attention-Based Knowledge Distillation"
_ICLR.cc/2024/Conference — ICLR 2024 Conference Withdrawn Submission_

### Official Review · Reviewer_E3QF · 2023-10-28

**Soundness:** 2 fair
**Presentation:** 3 good
**Contribution:** 2 fair
**Rating:** 5
**Confidence:** 4

**Summary:**

This paper presents a method for distilling GNNs to improve inference speed. The key idea is to not only distill the final representations but also the intermediate layers. A key challenge is what layer pair between teacher and student should be chosen for the distillation. To address these challenges, the authors propose to use an attention map to calculate the weights. Experiments on OGBN-Mag and OGBN-Arxiv show that the proposed method outperforms the baselines. The method also shows advantages on other datasets such as Cora, Citeseer, Pubmed, and NELL with GCNII as the teacher.

**Strengths:**

1. GNN distillation is an important problem to enable GNNs to be applied to many real-world problems.
2. The proposed method outperforms the baselines on two large OGBN datasets, OGBN-Mag, and  OGBN-Arxiv.
3. Abalation studies have been presented to help understand the method.

**Weaknesses:**

1. The idea of distilling hidden layers is very novel and is very common in knowledge distillation. This method also does not consider the graph structure in distillation.
2. GNNs typically are not deep. So it is unclear how much a small attention can contribute to the performance. We can also use some simple heuristics to identify the pair, such as aligning later layers with later layers and beginning layers with beginning layers.
3. The distillation process could be costly as the method distills every teacher-student pair.
4. I am curious how the method performs on other OGBN datasets, such as OGBN-product.

**Questions:**

See weaknesses.

---

### Official Review · Reviewer_GJj1 · 2023-10-31

**Soundness:** 2 fair
**Presentation:** 2 fair
**Contribution:** 2 fair
**Rating:** 3
**Confidence:** 3

**Summary:**

This paper proposes Attention-Based Knowledge Distillation (ABKD), a method to compress large graph neural networks (GNNs) into smaller networks using knowledge distillation. ABKD uses attention to identify important intermediate layer pairs between teacher and student models, and aligns the outputs of these layers. Experiments are conducted on OGB graph datasets using GCN, GAT, and GraphSAGE models. Results show 1-2% accuracy gains over prior distillation methods.
Overall the paper is marginally below the acceptance bar for ICLR. More extensive experiments and analysis would be needed to demonstrate the advantages of this approach over prior graph distillation methods. I would recommend rejection of the paper in its current form.

**Strengths:**

- Tackles an interesting problem - compressing GNNs for lower latency and memory usage.
- Provides both theoretical analysis and empirical results to validate the approach.

**Weaknesses:**

- The accuracy improvements over prior methods are quite small, just 1-2% on relatively small datasets. The gains may not be significant enough to demonstrate a strong advantage.
- Ablation studies are limited to analyzing the impact of different components. More in-depth empirical analysis could be beneficial.
- Novelty is somewhat limited since attention to layer alignment exists in other domains. Adaptations for graphs are incremental.

**Questions:**

- Have the authors experimented with larger graph datasets? The improvements appear quite modest on the small OGB datasets tested.
- Is an attention mechanism the best way to identify important layer pairs? Or could simpler approaches like greedy selection work?
- More rigorous theory about why intermediate alignments are beneficial for GNNs could strengthen the novelty.

---

### Official Review · Reviewer_Vwgr · 2023-11-04

**Soundness:** 2 fair
**Presentation:** 2 fair
**Contribution:** 3 good
**Rating:** 3
**Confidence:** 4

**Summary:**

The paper revolves around knowledge distillation (KD) for Graph Neural Networks (GNNs). Prior approaches only align the last layer embeddings of the teacher and student models. The aim of this paper is to achieve better distillation by considering the intermediate representations of the student and teacher GNNs.

To achieve the knowledge distillation between the two models, the authors use an auxiliary loss function. To compute the loss function they calculate 1) an attention matrix (row-normalised, i.e. a student layer can freely attend to more than one teacher layer, but not the inverse) representing the importance of teacher layers to student layers and 2) a dissimilarity matrix, equal to the difference of the graph embeddings at different layers between the teacher and student models. The loss aims to minimise the row-wise mean of $A \odot B$.

All experiments are performed on transductive datasets (authors, feel free to correct me if I misunderstood). From my paper understanding -- all results are on node classification tasks (rather than, e.g. link-prediction). GNN models used are RGCN/GAT/GraphSage/GCNII. A comparison to other knowledge distillation approaches is also provided.

Results suggest that the proposed method outperforms **most** prior knowledge distillation methods and a number of ablations are provided.

**Strengths:**

* Novel idea originating from the fact past methods do not take into account intermediate representations
* Simple and intuitive method
* Visualisations of the attention/dissimilarity maps.

**Weaknesses:**

* Presentation/typesetting is poor and can be improved

  **Typesetting:** I feel authors have not adhered to the formatting instructions (e.g. using different bibliography style, table captions under tables, smaller font size for figure and table captions). Matrices and number sets use the same formatting family (e.g. not using $\mathbb{R}$, but $R$ for reals, etc.). Table formatting can also be improved [link1](https://people.inf.ethz.ch/markusp/teaching/guides/guide-tables.pdf)

  **Presentation:** Table 1 feels unnecessary. $\S$3.1.3 defines only GCN formulation, but experiments use RGCN/GAT/GraphSage (why was the formulation included then?) Proofs and other mathematical details are not easy to follow. As a simple example, you can express the dissimilarity score $D_{ij}$ by first defining graph feature vectors for layers $i$ and $j$ of teacher and student (by, e.g. using $\sum$ operator) and then simply defining $D_{ij}$ as the Euclidean distance between the difference of those two feature vectors. Model format in Tables 2/3 is not provided.
* Although the approach is interesting, it considers pooled graph embeddings, and not aligning individual node embeddings (as in G-CRP).
* Please check equations carefully, as I am not confident they always match the code. E.g. in eq. 4 you calculate the squared Euclidean distance between graph feature vectors and then normalise it in eq.5, but in the code, you seem to normalise it trice: L170+L171 and L181 of `distillation.py`.
* Theorem proving is not formal. If you want to formalise theorem 1, please define "will be affected" mathematically and prove it.
* There are some experimental details missing: Learning rate, number of epochs, etc. README in the repository uses quite cryptic flag names to infer hyperparameter setup easily.
* Hardware is reported in the paper, but I found no runtimes reported in the main text or supplementary.
* Some experiments do not seem very sensible: Although past literature has studied knowledge distillation from deeper into a shallower network (e.g. 5 layer to 2 layer in G-CRD), performing distillation from a 64L network to a 4L network seems counterintuitive -- you are forcing the student GNN to capture properties of a 64-hop neighbourhood with a 4-hop filter. Even more surprisingly, according to Figure 5 **all** of the 64 layers' embeddings (each of dimensionality 64) get compressed into 1 layer embedding of dimensionality 4 for Pubmed. I believe the results here have an alternative explanation (e.g. something related to dataset complexity/diameter/etc).
* Some experimental results do not seem statistically significant: In Table 3, the difference to G-CRD is well within 2 standard deviations for 3/4 datasets. Similarly, in Table 6, projection vs no projection is quite close (could this be only due to the additional parameters used during the KD phase?)
* All experiments only cover transductive learning.

**Questions:**

* Does 4L-4H in brackets after model name mean that the hidden dimensionality is 4?
* How costly is ABKD compared to training the student model alone and compared to other distillation methods?
* How were No Training column's results obtained for Table 4?
* How do corresponding dissimilarity maps look like in Figure 5?
* The attention coefficient maps look too sharp (i.e. only 0s and 1s). Can you provide any intuition/justification why this happens?
* Are results statistically significant (e.g. for Tables 3/5/6)? Have you performed any significance testing?
* How does ABKD perform with inductive tasks (ones where we test the GNN on previously unseen graphs)? G-CRD give results with MOLHIV, for example.

---

### Official Review · Reviewer_51nj · 2023-11-05

**Soundness:** 2 fair
**Presentation:** 2 fair
**Contribution:** 2 fair
**Rating:** 3
**Confidence:** 3

**Summary:**

This work discusses a novel approach (ABKD) to compressing Graph Neural Networks (GNNs) to reduce latency issues. This compression is achieved through a method called Attention-Based Knowledge Distillation (ABKD), which aims to distill the knowledge of large GNNs into smaller ones. Unlike previous methods that primarily focus on the outputs of the last layers of GNNs, ABKD leverages attention mechanisms to identify and align important intermediate layers of teacher-student GNN pairs. The paper claims that ABKD achieves better compression ratios with a smaller drop in accuracy compared to state-of-the-art techniques.

**Strengths:**

The strengths of the paper can be summarized as follows:

**Attention Mechanism:** The use of attention to identify critical teacher-student layer pairs is an innovative feature that likely contributes to the method's effectiveness.

**Performance Improvement:** The paper claims an increase in accuracy and a significant compression ratio, indicating that the method does not sacrifice performance for efficiency.

**Applicability to Large Datasets:** The method has been tested on a large graph dataset, OGBN-Mag, showcasing its potential for real-world applications.

**Weaknesses:**

This work proposes an Attention-Based Knowledge Distillation (ABKD) approach for the compression of Graph Neural Networks (GNNs). Despite the clear effort and potential of this work, there are several concerns to be addressed:

**Missing Existing Literature:**

The manuscript does not sufficiently contextualize the proposed ABKD within the landscape of existing research on knowledge distillation for GNNs. As the references provided [1-7] illustrate, distilling knowledge from intermediate layers of GNNs is not a novel concept. The differentiation of ABKD from these established methods is not clearly articulated. A detailed comparison with these specific works would be crucial to highlight the unique contributions of ABKD.

**Technical Novelty and Justification:**

The paper would benefit from a clearer technical exposition of ABKD, especially regarding how it improves upon the limitations of the methods cited. There should be a specific discussion on the advantages that ABKD offers over these prior works, supported by empirical evidence. The proposed attention-based KD introduces additional learnable parameters. It is not clear how the supervision prevents convergence to trivial solutions.

**Empirical Comparison and Evaluation:**

The evaluation lacks a comprehensive comparison with the methods listed in the references. Including such comparisons would provide a clearer picture of the performance and efficiency gains of ABKD. We recommend an expanded evaluation section that not only benchmarks against these methods but also discusses the reasons for any observed improvements.

**Clarity of Contribution:**

The current contribution of the paper is overshadowed by the lack of distinction from prior works. It is crucial to clarify what aspects of ABKD are novel and how they advance the state of the art.


1. Kim, J., Jung, J. and Kang, U., 2021. Compressing deep graph convolution network with multi-staged knowledge distillation. Plos one, 16(8), p.e0256187.

2. Jing, Y., Yang, Y., Wang, X., Song, M. and Tao, D., 2021. Amalgamating knowledge from heterogeneous graph neural networks. In Proceedings of the IEEE/CVF conference on computer vision and pattern recognition (pp. 15709-15718).

3. Yang, C., Liu, J. and Shi, C., 2021, April. Extract the knowledge of graph neural networks and go beyond it: An effective knowledge distillation framework. In Proceedings of the web conference 2021 (pp. 1227-1237).

4. Zheng, W., Huang, E.W., Rao, N., Katariya, S., Wang, Z. and Subbian, K., 2021. Cold brew: Distilling graph node representations with incomplete or missing neighborhoods. arXiv preprint arXiv:2111.04840.

5. Wang, C., Wang, Z., Chen, D., Zhou, S., Feng, Y. and Chen, C., 2021. Online adversarial distillation for graph neural networks. arXiv preprint arXiv:2112.13966.

6. Joshi, C.K., Liu, F., Xun, X., Lin, J. and Foo, C.S., 2022. On representation knowledge distillation for graph neural networks. IEEE Transactions on Neural Networks and Learning Systems.

7. Liu, J., Zheng, T. and Hao, Q., 2022. HIRE: Distilling high-order relational knowledge from heterogeneous graph neural networks. Neurocomputing, 507, pp.67-83.

**Questions:**

See weaknesses.